# Stability of Graph Scattering Transforms

**Fernando Gama**
Dept. of Electrical and Systems Eng.
University of Pennsylvania
Philadelphia, PA 19104
fgama@seas.upenn.edu

**Joan Bruna**
Courant Institute of Math. Sci.
New York University
New York, NY 10012
bruna@cims.nyu.edu

**Alejandro Ribeiro**
Dept. of Electrical and Systems Eng.
University of Pennsylvania
Philadelphia, PA 19104
aribeiro@seas.upenn.edu

## Abstract

Scattering transforms are non-trainable deep convolutional architectures that exploit the multi-scale resolution of a wavelet filter bank to obtain an appropriate representation of data. More importantly, they are proven invariant to translations, and stable to perturbations that are close to translations. This stability property provides the scattering transform with a robustness to small changes in the metric domain of the data. When considering network data, regular convolutions do not hold since the data domain presents an irregular structure given by the network topology.

In this work, we extend scattering transforms to network data by using multiresolution graph wavelets, whose computation can be obtained by means of graph convolutions. Furthermore, we prove that the resulting graph scattering transforms are stable to metric perturbations of the underlying network. This renders graph scattering transforms robust to changes on the network topology, making it particularly useful for cases of transfer learning, topology estimation or time-varying graphs.

## 1 Introduction

Linear information processing architectures have been the preferred tool for extracting useful information from data due to their robustness and provable performance [1–6]. With the desire to model increasingly more complex mappings between data and useful information, linear approaches started to fall short in terms of performance, giving rise to a myriad of other nonlinear alternatives [2, Chap. 8], [6, Part 4]. Of these, arguably the most successful have been convolutional neural networks (CNNs) [7]. CNNs consist of a cascade of layers, each of which computes a convolution with a bank of filters followed by a pointwise nonlinearity, and act as a parameterization of the nonlinear mapping between the input data and the desired useful information [8].

The inclusion of nonlinearities coupled with the use of trained coefficients has effectively increased the performance, but it also has obscured the limits and guarantees of CNNs [9]. In the theoretical realm, [10, 11] opted for controlling for one of the sources of uncertainty, by fixing the bank of filters to be a set of pre-defined, multiresolution wavelets. Then, [10] proved that under admissible conditions on the wavelets, the resulting non-trainable CNN (called scattering transform) satisfies energy conservation, as well as stability to domain deformations that are close to translations. In

essence, the stability properties of non-trainable scattering transforms constitutes one of the main theoretical results explaining the success of CNNs.

Data stemming from networks, however, does not exhibit a regular inherent structure that can be effectively exploited by convolutions. Data elements are, instead, related by arbitrary pairwise relationships described by an underlying graph support. Graph neural networks (GNNs) have emerged as successful architectures that exploit this graph structure [12–15]. GNNs, mimicking the overall architecture of CNNs, also consist of a cascade of layers, but constrain the linear transform in each layer to be a graph convolution with a bank of graph filters [16–20]. Graph convolutions are, in analogy with traditional (regular) convolutions, a weighted sum of shifted versions of the input signal. The filter taps (weights) of the bank of graph filters are also obtained by minimizing a cost function over the training set. The mathematical challenges arising from the use of trainable filters and pointwise nonlinearities have prevented a rapid development of the theory of GNNs as well. Moreover, the particularities of the underlying irregular structure supporting network data raises challenges of its own.

Following the roadmap of the Euclidean, regular case, in this paper we pursue the investigation of the benefits of GNN architectures through the lens of their non-trainable counterparts, where filters are designed from multiresolution wavelet families. Several papers [21–23] have made initial progress in defining scattering graph representation and studying their stability properties with respect to metric deformations of the domain. However, most of these results offer bounds that depend on the graph topology and do not hold for certain graphs or when graphs are very large. Additionally, these works do not recover the Euclidean scattering stability result on Euclidean grids. The main theoretical contribution of this work is to establish stability to *relative* metric deformations for a wide class of graph wavelet families, yielding a bound that is independent on the graph topology (it only depends on the size of the deformation and the representation architecture).

The rest of the paper is structured as follows. In section 2 we discuss related works. In section 3 we define the scattering transform architecture, use the graph signal processing framework to describe network data (Sec. 3.1), and define graph scattering transforms (GSTs) using graph wavelets (Sec. 3.2). Then, we proceed to prove our main theoretical claims in section 4. Namely, that GSTs are permutation invariant (Prop. 1), and that they are stable (Theorem 1) under a relative perturbation model (Sec. 4.1). Finally, we show through numerical experiments in section 5, that the GST representation is not only stable, but also captures rich enough information. Conclusions are drawn in section 6.

## 2  Related Work

The particular property of stability has been investigated, in analogy to scattering transforms, for the case of non-trainable graph wavelet filter banks [21, 22]. More specifically, [21] studies the stability of graph scattering transforms to permutations, as well as to perturbations on the eigenvalues and eigenvectors of the underlying graph support. Furthermore, [21] derives results on energy conservation. The bounds obtained on approximate permutation invariance grow with the size of the graph, while the bounds on the stability to graph perturbations are applicable only for changes in edge weights that are smaller with increasing graph size (i.e. larger graphs admit smaller edge weight changes). Alternatively, in [22], graph scattering transforms using diffusion wavelets [24] are considered. Perturbations are defined in terms of changes in the underlying graph support, and measured using diffusion distances [25,26]. The bound obtained on the output for different underlying graph supports, depends on the spectral gap of the filter, making this bound quite loose in some cases [22]. We note that [27] isolates the bound on the powers of the graph shift operator [22, eq. (23)] and generalizes it for arbitrary graph filters. As such, the resulting bound also depends on the spectral gap. Finally, we draw attention to the work in [28]. This work defines geometric scattering transforms, which are an extension of diffusion scattering [22], by using a lazy random walk adjacency as the matrix representation and considering higher-order moments for the low-pass operator. Furthermore, they do an exhaustive experimental comparison between geometric scattering transforms and a myriad of graph-based machine learning techniques.

# 3 Graph scattering transforms

A scattering transform network [10, 11] is a deep convolutional architecture comprised of three basic elements: (i) a bank of multiresolution wavelets $\{\mathbf{h}_j\}_{j=1}^J$, (ii) a pointwise nonlinearity $\rho$ (absolute value), and (iii) a low-pass average operator $U$. These elements are combined sequentially to produce a representation $\mathbf{\Phi}(\mathbf{x})$ of the data $\mathbf{x}$. More specifically, as illustrated in Fig. 1, each of the $J$ wavelets is applied to each of the *nodes* of the previous layer, generating $J$ new *nodes* to which the nonlinearity is applied. The output is harvested at each *node* by computing a low-pass average through the operator $U$. For a scattering transform with $L$ layers, the number of coefficients of the representation $\mathbf{\Phi}(\mathbf{x})$ is $\sum_{\ell=0}^{L-1} J^\ell = (J^L - 1)/(J - 1)$, independent of the size of the input data.

Each coefficient of the scattering transform is determined by the sequence of wavelet indices (resolution scales) traversed to compute it. We call this sequence a *path*. Let $\mathcal{J}(\ell) = \{1, \ldots, J\}^\ell$ be a shorthand for the space of all possible $\ell$-tuples with $J$ elements, defined for all $\ell > 0$ and where we set $\mathcal{J}(0) = \{0\}$. Then, we can define the path $p_j(\ell) : \mathbb{N} \to \mathcal{J}(\ell)$ as the mapping between $j \in \mathbb{N}$ and the specific sequence $p_j(\ell) = (j_1, \ldots, j_\ell)$ of length $\ell$ comprised of a combination of indices from 1 to $J$ (tuples), with $p_1(0) = 0$. Sequences $p_j(\ell)$ and $p_i(\ell)$ are distinct for $j \neq i$ so that $\{p_j(\ell)\}_{j=1,\ldots,J^\ell} \equiv \mathcal{J}(\ell)$ is the space of all possible tuples. We denote by $\mathcal{J}(\mathcal{L}) = \{p_j(\ell) \in \mathcal{J}(\ell), \forall\, j \in \{1, \ldots, J^\ell\}, \forall\, \ell \in \{0, \ldots, L-1\}\}$ the set of all sequences for all values of $\ell$, see Fig. 1.

With this notation in place, the scattering transform $\mathbf{\Phi}(\mathbf{x})$ of the data $\mathbf{x}$ is the collection of *scattering coefficients* $\phi_{p_j(\ell)}(\mathbf{x})$

$$\mathbf{\Phi}(\mathbf{x}) = \left\{\phi_{p_j(\ell)}(\mathbf{x})\right\}_{\mathcal{J}(\mathcal{L})} := \left\{\phi_{p_j(\ell)}(\mathbf{x})\right\}_{p_j(\ell) \in \mathcal{J}(\ell), \ell=0,\ldots,L-1}. \tag{1}$$

For a given sequence $p_j(\ell) = (j_1, \ldots, j_\ell) \in \mathcal{J}(\ell)$, the scattering coefficient $\phi_{p_j(\ell)}$ is computed as

$$\phi_{p_j(\ell)}(\mathbf{x}) = U\left[(\rho \mathbf{h}_j)_{p_j(\ell)} * \mathbf{x}\right] = U\mathbf{x}_{p_j(\ell)} \tag{2}$$

where the notation $[(\rho \mathbf{h}_j)_{p_j(\ell)} * \mathbf{x}] := [(\rho \mathbf{h}_j)_{j \in p_j(\ell)} * \mathbf{x}] = \rho \mathbf{h}_{j_\ell} * \cdots * \rho \mathbf{h}_{j_1} * \mathbf{x}$ is a shorthand for the repeated application of pointwise nonlinearities $\rho$ and wavelets $\mathbf{h}_j$ following the scale indices determined by the path $p_j(\ell)$. The operator $U$ outputs as a scalar, computed by means of a *summarizing* low-pass linear operator, typically an average or a sum. Note that we set $\phi_{p_1(0)} = \phi_0 = U\mathbf{x}$. The energy of the scattering transform is given by the energy in its coefficients

$$\|\mathbf{\Phi}(\mathbf{x})\|^2 = \sum_{\mathcal{J}(\mathcal{L})} |\phi_{p_j(\ell)}(\mathbf{x})|^2 = \sum_{\ell=0}^{L-1} \sum_{j=1}^{J^\ell} |\phi_{p_j(\ell)}(\mathbf{x})|^2. \tag{3}$$

## 3.1 Network data

The scattering transform relies heavily on the use of the convolution to filter the data through the wavelet multiresolution bank. The convolution operation, in turn, depends on the data exhibiting a regular structure, such that contiguous data elements represent elements that are spatially or temporally related. This is not the case for network data, whereby data elements are related by arbitrary pairwise relationships determined by the underlying network topology.

To describe network data, we denote by $\mathcal{G} = (\mathcal{V}, \mathcal{E}, \mathcal{W})$ the underlying graph support, with $\mathcal{V}$ the set of $N$ nodes, $\mathcal{E} \subseteq \mathcal{V} \times \mathcal{V}$ the set of edges, and $\mathcal{W} : \mathcal{E} \to \mathbb{R}$ the edge weighing function. The data $\mathbf{x} \in \mathbb{R}^N$ is modeled as a *graph signal* where each element $[\mathbf{x}]_i = x_i$ is the value of the data at node $i \in \mathcal{V}$[1] [15]. To operationally relate data $\mathbf{x}$ with the underlying graph support $\mathcal{G}$, we define a *graph shift operator* (GSO) $\mathbf{S} \in \mathbb{R}^{N \times N}$ which is a matrix representation of the graph that respects its sparsity, i.e. $[\mathbf{S}]_{ij} = s_{ij}$ can be nonzero, only if $(j, i) \in \mathcal{E}$ or if $i = j$ [15]. Examples of GSOs commonly used in the literature include the adjacency matrix [12, 13], the Laplacian matrix [14], and their normalized counterparts [18, 22].

The operation $\mathbf{Sx}$ is, due to the sparsity constraint of $\mathbf{S}$, a local, linear operation, by which each node $i$ in the network updates its value by means of a weighted linear combination of the signal values at

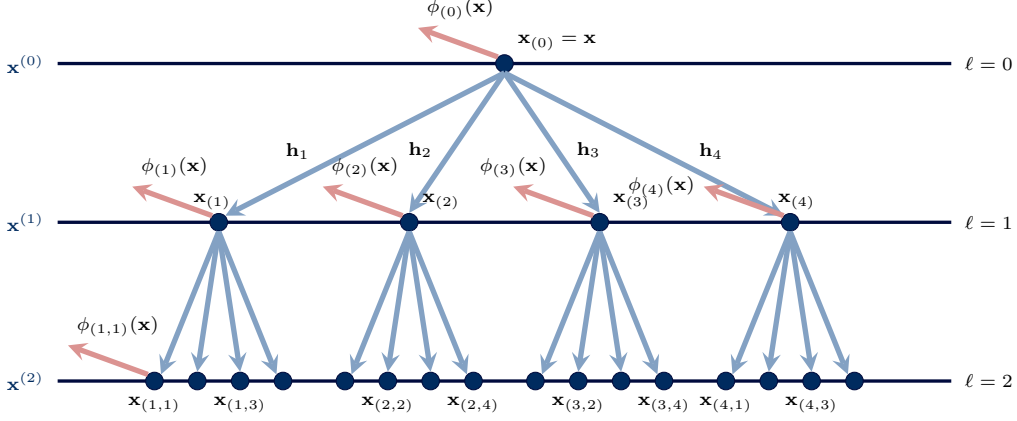

Figure 1. Graph scattering transform. Illustration for $J = 4$ scales and $L = 3$ layers. At layer $\ell = 0$ we have a single coefficient $\phi_{(0)}(\mathbf{x})$ since $\mathcal{J}(0) = \{0\}$, which is obtained by applying the low-pass operator $U$ to the input data $\mathbf{x}$ directly. In the next layer $\ell = 1$ we have $J^1 = 4$ coefficients. We generate 4 *nodes* by applying each of the 4 wavelets $\mathbf{h}_j$ to the input data followed by a pointwise nonlinearity, yielding $\mathbf{x}_{p_j(1)}$ where $\mathcal{J}(1) = \{1, 2, 3, 4\}$. Then, we obtain the output coefficients $\phi_{p_j(\ell)}(\mathbf{x})$ by means of the low-pass operator $U$. For the following layer $\ell = 2$ we have $J^2 = 16$ coefficients. For each of the $J$ previous nodes, we apply each of the wavelets yielding $J$ new nodes for each one of them, followed by the nonlinearity $\rho$. Then, we obtain the new 16 coefficients by applying the low-pass operator $U$.

neighboring nodes $j \in \mathcal{N}_i$

$$[\mathbf{Sx}]_i = \sum_{j \in \mathcal{N}_i} s_{ij} x_j. \tag{4}$$

Note that, while $\mathbf{Sx}$ computes a summary of the information in the one-hop neighborhood of each node, repeated application of $\mathbf{S}$ computes summaries from farther away neighborhoods, i.e. $\mathbf{S}^k \mathbf{x} = \mathbf{S}(\mathbf{S}^{k-1}\mathbf{x})$ computes a summary from the $k$-hop neighborhood. This allows for the definition of *graph convolutions*, in analogy with regular convolutions. More precisely, since regular convolutions are linear combinations of data that is spatially or temporally nearby, graph convolutions are defined as a linear combination of data located at consecutive neighborhoods

$$\mathbf{h} *_{\mathbf{S}} \mathbf{x} = \sum_{k=0}^{K-1} h_k \mathbf{S}^k \mathbf{x} = \mathbf{H}(\mathbf{S})\mathbf{x} \tag{5}$$

where $\mathbf{h} = \{h_0, \ldots, h_{K-1}\}$ is the set of $K$ filter coefficients, and where we use $*_{\mathbf{S}}$ to denote a graph convolution over GSO $\mathbf{S}$ [29]. We note that the output of the graph convolution is another graph signal defined over the same graph $\mathcal{G}$ as the input $\mathbf{x}$.

The graph convolution (5) also satisfies the convolution theorem [30, Sec. 2.9.6], which states that convolution implies multiplication in frequency domain. We define the *graph frequency domain* in terms of the eigendecomposition of the GSO, which we assume to be normal $\mathbf{S} = \mathbf{V}\boldsymbol{\Lambda}\mathbf{V}^{\mathsf{H}}$, where $\mathbf{V}$ is the matrix of eigenvectors which determines the frequency basis signals, and $\boldsymbol{\Lambda}$ is the diagonal matrix of eigenvalues that determines the frequency coefficients [13]. The graph Fourier transform (GFT) of a graph signal is defined as the projection of the graph signal onto the space of frequency basis signals $\tilde{\mathbf{x}} = \mathbf{V}^{\mathsf{H}}\mathbf{x}$. So, if we compute the GFT of the output of the graph convolution, we get

$$\tilde{\mathbf{y}} = \mathbf{V}^{\mathsf{H}}\mathbf{y} = \mathbf{V}^{\mathsf{H}}(\mathbf{h} *_{\mathbf{S}} \mathbf{x}) = \mathbf{V}^{\mathsf{H}}\sum_{k=0}^{K-1} h_k \mathbf{S}^k \mathbf{x} = \sum_{k=0}^{K-1} h_k \boldsymbol{\Lambda}^k \tilde{\mathbf{x}} = \mathrm{diag}(\tilde{\mathbf{h}})\tilde{\mathbf{x}} = \tilde{\mathbf{h}} \circ \tilde{\mathbf{x}} \tag{6}$$

where $\circ$ denotes the elementwise (Hadamard) product, yielding an multiplication of the GFT of the filter taps with the GFT of the signal. We note that the GFT $\tilde{\mathbf{h}}$ of the filter coefficients $\mathbf{h}$ is given by a polynomial on the eigenvalues of the graph

$$[\tilde{\mathbf{h}}]_i = \tilde{h}_i = h(\lambda_i) \quad \text{with} \quad h(\lambda) = \sum_{k=0}^{K-1} h_k \lambda^k. \tag{7}$$

It is very interesting to remark that the GFT of the filter is characterized by the same function $h(\lambda)$, which depends on the filter coefficients, irrespective of the graph. The specific value of the frequency coefficients of the filter (and its impact on the output), however, is obtained by instantiating $h(\lambda)$ on the eigenvalues of the given graph. But $h(\lambda)$ still characterizes the GFT of the filter taps for all graphs.

## 3.2 Graph wavelets and graph scattering transforms

Graph wavelets are typically defined in the graph frequency domain, by specifying a specific form on the function $h(\lambda)$ [31, 32]. For instance, [31] proposes to choose a mother wavelet (wave generating kernel) $h(\lambda)$ from the regular Wavelet literature and then construct all the rest of the wavelet scales by rescaling the continuous parameter $\lambda$ before sampling it with the eigenvalues corresponding to the specific graph, see [31, eq. (65)] for a concrete example of a graph wavelet. This same construction method is further developed in [32] to obtain graph wavelets that are adapted to the spectrum (i.e. that localize the wavelets around the actual eigenvalues of the given graph, instead of just sampling rescaled versions of the wavelets). Concrete examples of graph wavelets are given in [32, Sec. IV-A].

Once the multiresolution wavelet filter bank is defined $\{h_j(\lambda)\}_{j=1}^{J}$ we proceed to compute the output by filtering each graph signal with the corresponding wavelet on the given graph. More precisely, consider $\mathbf{S} = \mathbf{V}\mathbf{\Lambda}\mathbf{V}^{\mathsf{H}}$ and define $\tilde{\mathbf{h}}_j = [h_j(\lambda_1), \dots, h_j(\lambda_N)]^{\mathsf{T}}$ by evaluating $h_j(\lambda)$ on each of the $N$ eigenvalues of $\mathbf{S}$. Then, we obtain [cf. (6)]

$$\mathbf{y}_j = \mathbf{V}\tilde{\mathbf{y}}_j = \mathbf{V}\text{diag}(\tilde{\mathbf{h}}_j)\tilde{\mathbf{x}} = \mathbf{V}\text{diag}(\tilde{\mathbf{h}}_j)\mathbf{V}^{\mathsf{H}}\mathbf{x} = \mathbf{H}_j(\mathbf{S})\mathbf{x} \tag{8}$$

where the output $\mathbf{y}_j$ for each scale is computed as a linear operation $\mathbf{H}_j(\mathbf{S})$ on the input data $\mathbf{x}$.

An important property of wavelets in general, and graph wavelets in particular, is that they conform a frame [32]. This controls the spread of energy when computing the multiresolution output. For $0 < A \le B < \infty$ and a multiresolution wavelet bank $\{\mathbf{h}_j\}_{j=1}^{J}$, it conforms a frame if

$$A^2\|\mathbf{x}\|^2 \le \sum_{j=1}^{J} \|\mathbf{H}_j(\mathbf{S})\mathbf{x}\|^2 \le B^2\|\mathbf{x}\|^2. \tag{9}$$

For wavelets constructed following the above method, it is proven that they always conform a frame [31, Theorem 5.6]. In particular, the work in [32] designs graph wavelets that are tight, which means that $A = B$ in (9).

We note that every analytic function $h(\lambda)$ can be computed in terms of a graph convolution (5). More precisely, an analytic function can be written in terms of a power series, but since graphs are finite, in virtue of the Cayley-Hamilton theorem [33, Theorem 2.4.2], this power series can be written as a polynomial of degree at most $N - 1$, i.e. by setting $K = N$ in (5). Moreover, [31, Sec. 6] provides a method for fast computation of the output of graph wavelets, by approximation with a polynomial of order $K \ll N$.

Finally, we define a *graph scattering transform* (GST), as an architecture of the form (1)-(2), but where we replace regular convolutions by graph convolutions (5) with a bank of analytic graph wavelets $\{\mathbf{h}_j\}_{j=1}^{J}$ that conform a frame (9).

## 4 Stability to perturbations

Regular scattering transforms have been proven invariant to translations and stable to perturbations (or deformations) that are close to translations. That is, the difference on the scattering transform of the original data and that of the perturbed data, is proportional to the size of the perturbation. In the case of network data, we consider perturbations to the underlying graph support. More specifically, we consider a $N$-node graph $\mathcal{G}$ with a GSO $\mathbf{S}$ and a *perturbed* $N$-node graph $\hat{\mathcal{G}}$ with a GSO $\hat{\mathbf{S}}$. The objective, then, is to prove that the GST is a stable operation under such perturbations, namely that

$$\left\|\mathbf{\Phi}(\mathbf{S}, \mathbf{x}) - \mathbf{\Phi}(\hat{\mathbf{S}}, \mathbf{x})\right\| \lesssim d(\mathbf{S}, \hat{\mathbf{S}}) \tag{10}$$

for some distance $d(\mathbf{S}, \hat{\mathbf{S}})$ measuring the size of the perturbation. Perturbations on the underlying graph support are particularly useful in cases when the graph is unknown and needs to be estimated

[34], or when the graph changes with time [35]. Note that, since the wavelet functions $h_j(\lambda)$ are fixed by design, then the analysis centers around how changes in the underlying graph support affect the eigenvalues which instantiate the GFT of the wavelets, and how does the function $h_j(\lambda)$ change its output when instantiated in different eigenvalues.

First, we consider perturbations that arise from permutations, that amount to node reorderings. Define the set of permutation matrices as

$$\mathcal{P} = \left\{ \mathbf{P} \in \{0,1\}^{N \times N} : \mathbf{P1} = \mathbf{1} \, , \, \mathbf{P}^\mathsf{T}\mathbf{1} = \mathbf{1} \right\}. \tag{11}$$

Next, we show that the GST is invariant to permutations

**Proposition 1** (Permutation invariance)**.** *Let $\mathcal{G}$ be a graph with a GSO $\mathbf{S}$, and let $\widehat{\mathcal{G}}$ be a permuted graph with GSO $\hat{\mathbf{S}} = \mathbf{P}^\mathsf{T}\mathbf{SP}$. Let $\mathbf{x}$ be the input data and $\hat{\mathbf{x}} = \mathbf{P}^\mathsf{T}\mathbf{x}$ the correspondingly permuted data. Then, it holds that*

$$\mathbf{\Phi}(\mathbf{S}, \mathbf{x}) = \mathbf{\Phi}(\hat{\mathbf{S}}, \hat{\mathbf{x}}) \tag{12}$$

Prop. 1 essentially states that the GST is independent of the chosen node ordering. Furthermore, it states that the GST exploits the topological symmetries present in the graph, i.e., that nodes with the same topological neighborhood yield the same output (if the value of the signal in the neighborhood is the same). In other words, different parts of the graph are distinct inasmuch as their neighborhood topologies are distinct.

## 4.1 Perturbation model

When considering arbitrary perturbations $\hat{\mathbf{S}}$ of $\mathbf{S}$, and in light of Prop. 1, we need to define a distance $d(\mathbf{S}, \hat{\mathbf{S}})$ such that, when $\hat{\mathbf{S}}$ is a permutation of $\mathbf{S}$, then $d(\mathbf{S}, \hat{\mathbf{S}}) = 0$. This would imply that, in the same way regular scattering transforms are invariant to translations and stable to perturbations that are close to translations, GSTs are invariant to permutations and stable to perturbations that are close to permutations. Define the set of permutations that make $\mathbf{S}$ and $\hat{\mathbf{S}}$ the closest as

$$\mathcal{P}_0 = \operatorname*{argmin}_{\mathbf{P} \in \mathcal{P}} \left\| \mathbf{P}^\mathsf{T}\hat{\mathbf{S}}\mathbf{P} - \mathbf{S} \right\|. \tag{13}$$

Then, we consider the set of error matrices to be

$$\mathcal{E}(\mathbf{S}, \hat{\mathbf{S}}) = \left\{ \mathbf{P}^\mathsf{T}\hat{\mathbf{S}}\mathbf{P} - \mathbf{S} = \mathbf{E}^\mathsf{H}\mathbf{S} + \mathbf{SE} \, , \, \mathbf{P} \in \mathcal{P}_0 \right\}. \tag{14}$$

And, since matrices $\mathbf{E} \in \mathcal{E}(\mathbf{S}, \hat{\mathbf{S}})$ measure the (relative) difference between $\mathbf{S}$ and $\hat{\mathbf{S}}$ accounting for all possible permutations, then we can define the distance that we use to measure perturbations as

$$d(\mathbf{S}, \hat{\mathbf{S}}) = \min_{\mathbf{E} \in \mathcal{E}(\mathbf{S}, \hat{\mathbf{S}})} \|\mathbf{E}\|. \tag{15}$$

Note that, indeed, if $\hat{\mathbf{S}} = \mathbf{P}^\mathsf{T}\mathbf{SP}$ is simply a permutation of $\mathbf{S}$, then $d(\mathbf{S}, \hat{\mathbf{S}}) = 0$.

**Remark 1.** The perturbation model in (14) and the consequent distance in (15) is a *relative* perturbation model. Relative perturbations successfully take into account structural characteristics of the underlying graph such as sparsity, average degree, or mean edge weights. This is not the case when considering absolute perturbations, which is the model adopted in [21, 22, 27].

## 4.2 Stability of graph wavelets

Changes in the underlying graph support directly affect the output of filtering the signal with a wavelet. That is, by changing the eigenvalues $\lambda_i$ on which the wavelet $h(\lambda)$ is instantiated, the filter taps $\tilde{h}_i$ are changed, and so does the output $\tilde{y}_i$ in virtue of (6). Thus, the first necessary result is to quantify the change in the output of a wavelet filter. Given a wavelet function $h(\lambda)$ and corresponding instantiations $\mathbf{H}(\mathbf{S})$ and $\mathbf{H}(\hat{\mathbf{S}})$, define the wavelet output difference as

$$\|\mathbf{H}(\mathbf{S}) - \mathbf{H}(\hat{\mathbf{S}})\| = \inf \left\{ c \geq 0 : \min_{\mathbf{P} \in \mathcal{P}} \left\| \mathbf{H}(\mathbf{S})\mathbf{x} - \mathbf{PH}(\mathbf{P}^\mathsf{T}\hat{\mathbf{S}}\mathbf{P})\mathbf{P}^\mathsf{T}\mathbf{x} \right\| \leq c\|\mathbf{x}\| \right\}. \tag{16}$$

We can then bound the wavelet output difference as shown next.

**Proposition 2** (Graph wavelet stability). *Let $\mathcal{G}$ be a graph with GSO $\mathbf{S}$ and $\widehat{\mathcal{G}}$ be the* perturbed *graph with GSO $\hat{\mathbf{S}}$, such that $d(\mathbf{S}, \hat{\mathbf{S}}) \leq \varepsilon/2$. Let $\mathbf{E} \in \mathcal{E}(\mathbf{S}, \hat{\mathbf{S}})$, consider its eigendecomposition $\mathbf{E} = \mathbf{U}\mathbf{M}\mathbf{U}^\mathsf{H}$ where the eigenvalues in $\mathbf{M} = \text{diag}(m_1, \dots, m_N)$ are ordered such that $|m_1| \leq \cdots \leq |m_N|$, and assume that the structural constraint $\|\mathbf{E}/m_N - \mathbf{I}\| \leq \varepsilon$ holds. Let $h(\lambda)$ be a graph wavelet that satisfies the integral Lipschitz constraint $|\lambda h'(\lambda)| \leq C$. Then, it holds that*

$$\|\mathbf{H}(\mathbf{S}) - \mathbf{H}(\hat{\mathbf{S}})\| \leq \varepsilon C + \mathcal{O}(\varepsilon^2) \tag{17}$$

The bound in Prop. 2 shows that the wavelet output difference is proportional to the size $\varepsilon$ of the perturbation. The structural constraint $\|\mathbf{E}/m_N - \mathbf{I}\|$ limits the changes in the structure of the graph, such as changes in sparsity or average degree and determines a cost for different perturbations. For instance, changing all the edge weights by the same amount does not affect the topology structure and thus $\|\mathbf{E}/m_N - \mathbf{I}\| = 0$. Also, while changing some edge weights by $\varepsilon/2$ satisfies the constraint, contracting some edges by $\varepsilon/2$ and dilating others in the same amount actually requires $\|\mathbf{E}/m_N - \mathbf{I}\| = \mathcal{O}(1)$. Finally, we note that graph perturbations such as adding and/or dropping edges altogether leads to $\|\mathbf{E}/m_N - \mathbf{I}\| = \mathcal{O}(1)$ as well. In a way, $d(\mathbf{S}, \hat{\mathbf{S}}) \leq \varepsilon/2$ limits the maximum edge weight change, while $\|\mathbf{E}/m_N - \mathbf{I}\| \leq \varepsilon$ limits how the edge weight changes affect the overall graph topology.

**Remark 2.** In what follows, we consider the low-pass average operator $U$ to be independent of the graph shift operator structure $\mathbf{S}$. In particular, we choose $U$ to be a straightforward average of the representation obtained at all nodes, i.e. $U = N^{-1}\mathbf{1}^\mathsf{T}$. In the appendix, we offer a proof of stability for cases in which $U$ depends on $\mathbf{S}$ as well.

### 4.3 Stability of graph scattering transform

The integral Lipschitz condition $|\lambda h'(\lambda)| \leq C$ requires the wavelet to be *constant* in high-eigenvalue frequencies (i.e. for $\lambda \to \infty$, the derivative $h'(\lambda)$ has to go to 0). This implies that information located in high-eigenvalue frequencies cannot be adequately discriminated (i.e. the output of the wavelet is the same for a broad band of the high-eigenvalue frequencies). Therefore, integral Lipschitz wavelets are stable, but not discriminative enough.

GSTs address this issue by incorporating pointwise nonlinearities. The effect of the pointwise nonlinearities is to cause a spillage of information throughout the frequency spectrum, in particular, into low-eigenvalue frequencies, which can then be discriminated in a stable fashion. Thus, GSTs are stable and discriminative information processing architectures.

To give a bound on the stability of the GST, we first derive a bound on the difference of a single GST coefficient, when computed on different graphs.

**Proposition 3** (GST coefficient stability). *Let $\mathcal{G}$ be a graph with GSO $\mathbf{S}$ and $\widehat{\mathcal{G}}$ be the* perturbed *graph with GSO $\hat{\mathbf{S}}$, such that $d(\mathbf{S}, \hat{\mathbf{S}}) \leq \varepsilon/2$. Let $\mathbf{E} \in \mathcal{E}(\mathbf{S}, \hat{\mathbf{S}})$, consider its eigendecomposition $\mathbf{E} = \mathbf{U}\mathbf{M}\mathbf{U}^\mathsf{H}$ where the eigenvalues in $\mathbf{M} = \text{diag}(m_1, \dots, m_N)$ are ordered such that $|m_1| \leq \cdots \leq |m_N|$, and assume that the structural constraint $\|\mathbf{E}/m_N - \mathbf{I}\| \leq \varepsilon$ holds. Consider a GST with $L$ layers and $J$ wavelet scales $h_j(\lambda)$, each of which satisfies the integral Lipschitz constraint $|\lambda h_j'(\lambda)| \leq C$ and conform a frame with bounds $0 < A \leq B$ [cf. (9)]. Then, for the coefficient $\phi_{p_j(\ell)}$ associated to path $p_j(\ell) = (j_1, \dots, j_\ell)$ it holds that*

$$|\phi_{p_j(\ell)}(\mathbf{S}, \mathbf{x}) - \phi_{p_j(\ell)}(\hat{\mathbf{S}}, \mathbf{x})| \leq \varepsilon C \ell B^{\ell-1} \|\mathbf{x}\|. \tag{18}$$

We note that for wavelets $h_j$ built as a rescaling of a mother wavelet $h$ [31], then it suffices for $h$ to satisfy the integral Lipschitz constraint $|\lambda h'(\lambda)| \leq C$ for all wavelets $h_j$ to satisfy the constraint as well. The bound in Prop. 3 can be used to prove stability for the entire GST representation.

**Theorem 1** (GST stability). *Under the conditions of Proposition 3 it holds that*

$$\left\|\boldsymbol{\Phi}(\mathbf{S}, \mathbf{x}) - \boldsymbol{\Phi}(\hat{\mathbf{S}}, \mathbf{x})\right\| \leq \frac{\varepsilon C}{B} \left(\sum_{\ell=0}^{L-1} \ell^2 (B^2 J)^\ell\right)^{1/2} \|\mathbf{x}\|. \tag{19}$$

First of all, we observe that the bound (19) is linear in the perturbation size $\varepsilon$, thus proving stability of the GST transform. Also, the proportionality constant depends on the characteristics of the GST

architecture, but not on the spectral gap nor any other characteristic of the underlying graph. It is linear also in the integral Lipschitz constant $C$, and depends exponentially on the upper bound of the filters $B$ and on the number of scales $J$, with the exponential factor given by the number of layers $L$.

Theorem 1 provides a bound that is independent of graph properties. This in contrast to results in [21, 22, 27] that depend on spectral signatures of the graph. An interesting consequence of this fact is that it makes it ready to take limits as we grow the number of nodes in the graph. There is, in fact, no limit to be taken as the bound holds for all graphs.

Of particular importance is the limit of a line graph in which case we partially recover the seminal stability results for scattering transforms using regular convolutions in [10]. The difference between Theorem 1 and the results in [10] is our restriction that the perturbation matrix be close to an identity. This means we can perturb the line graph by dilating all edges or by contracting all edges. Dilations and contractions can be different for different nodes but we cannot have a mix of dilation and contraction in different parts of the line. This is allowed in [10] where perturbations are arbitrary diffeomorphisms. Yet, we note that even in the context of diffeomorphisms, perturbations such as dropping an edge, still have a large gradient since it implies *folding* two points into one.

The reason for the relative weakness of the result is that [10] leverages extrinsic geometric information that is not available in an analysis that applies to arbitrary graphs. More precisely, [10] uses the knowledge of the underlying geometry of the Euclidean space to compute the bounds (i.e. they are derived for continuous space $\mathbb{R}^d$). In the context of this work, this amounts to using this extrinsic knowledge to bound the difference between the eigenvector basis of $\mathbf{S}$ and that of $\hat{\mathbf{S}}$. When we want a general result applicable to any graph, as is the case of Theorem 1, we need some (external) means of bounding how different the eigenvector basis are, and this is achieved by means of the structural constraint. All in all, this implies that if we have specific knowledge of the domain where the graph and the possible perturbations leave, then we can improve on (19) by leveraging this information to bound the difference in the eigenvector basis.

## 5 Numerical results

For the numerical experiments, we consider three scenarios[2]: representation error over a synthetic small world graph, authorship attribution and source localization over a Facebook subgraph, namely the same problems considered in [22]. We note that we are concerned with studying how changes in the underlying graph support $\mathbf{S}$ affect the output of the graph scattering transforms, when applied to the same input data $\mathbf{x}$. As such, we are interested in datasets where we can keep $\mathbf{x}$ constant while changing $\mathbf{S}$, i.e. scenarios involving data modeled as *graph signals*. In all cases, we study the GST carried out by three different wavelets: a monic cubic polynomial as suggested in [31], a tight Hann wavelet as in [32], and the geometric scattering introduced in [28]. For comparison, we consider the GFT as a linear, graph-based representation of the data and a trainable GIN [36]. We note that an exhaustive comparison between scattering transforms and other more traditional graph-based methods can be found in [28]. Complete details of all simulations are provided in the appendix. We consider GSTs with 6 scales and 3 layers, yielding representations with 43 coefficients when using the monic cubic polynomial [31] and a tight Hann wavelet [32]. For the geometric scattering we consider the low pass operator to compute 4 moments, as used in [28], leading to 172 coefficients. For scenarios two and three we consider a GFT with 43 coefficients and a GIN that produce 43 features in the hidden layer.

The first experiment is used to corroborate numerically the stability of the GST, and consists of computing the representation error obtained by transforming a white noise signal defined over a small world graph of 100 nodes. We compute the relative representation error $\|\mathbf{\Phi}(\mathbf{S}, \mathbf{x}) - \mathbf{\Phi}(\hat{\mathbf{S}}, \mathbf{x})\| / \|\mathbf{\Phi}(\mathbf{S}, \mathbf{x})\|$ and show the results in Fig. 2a. We observe that the GST incurs in up to 4 orders of magnitude less relative representation error than the GFT, resulting in markedly more stable representations. Within the different choices of wavelets, the geometric scattering is the more stable. Also, we show the theoretical bound of Theorem 1, computed as in (19) for the monic cubic polynomial wavelets, and where the values of $B$ and $C$ where obtained numerically (see the appendix for details). We see that the bound is not tight, but it is still lower than the GFT.

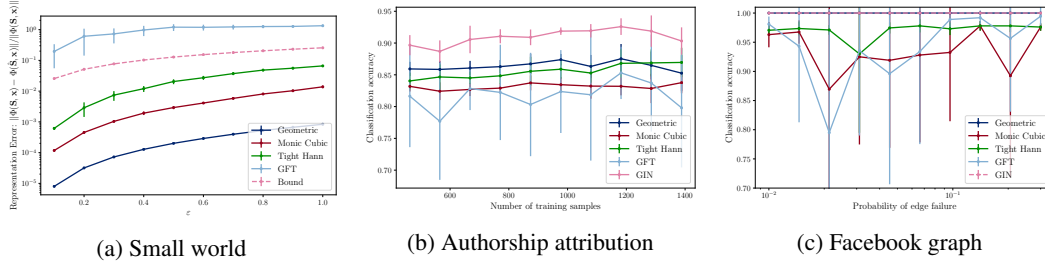

| (a) Small world | (b) Authorship attribution | (c) Facebook graph |

Figure 2. (a) Difference in representation between the signal defined using the original GSO $\mathbf{S}$ and using the GSO $\hat{\mathbf{S}}$ corresponding to the deformed graph as a function of the perturbation size $\varepsilon$ [cf. (15)]. (b)-(c) Classification accuracy as a function of perturbation for the authorship attribution and the Facebook graph, respectively.

For the second and third experiments, we consider two problems involving real-world data. The objective is twofold: (i) to show that the GST representations are, at least, as rich as the widely used GFT representation, and (ii) to consider stability to real-world perturbations (as opposed to controlled perturbations like in the first experiment). In Fig. 2b we show the classification accuracy in a problem involving authorship attribution of texts written by Jane Austen [37, 38], in the same scenario considered in [22]. The perturbation comes from considering different number of training excerpts and amounts to uncertainty in estimating the underlying graph topology. It is immediate to note that the performance obtained by a linear SVM classifier operating on the GST representation is comparable to that obtained when using the GFT, but worse than the GIN –which is understandable since the GIN has been trained for 40 epochs to fit the dataset–. We also observe that the oscillation of the mean classification accuracy of the GFT (as well as the large error bars) show that is is much less stable than the GST. In Fig. 2c we show the classification accuracy for a source localization problem over the 234-node Facebook subnetwork [39], as discussed in [22]. In this case, the perturbation comes from randomly dropping edges with probability given in the x-axis of the figure (from 0.01 to 0.3). We observe that the GST using tight Hann wavelets and the geometric scattering transform achieve better performance than the GFT and similar to that of the trained GIN, while the GST using monic cubic polynomials yields similar performance to the GFT. Finally, we note that the variability in the GFT is significantly larger than the geometric scattering and the tight Hann GST, but comparable to the Monic Cubic GST.

## 6    Conclusions

We have studied the stability properties of graph scattering transforms (GSTs) built with integral Lipschitz wavelets. We have introduced a relative perturbation model that takes into account the structure of the graph as well as its edge weights. We proved stability of the GST, by which changes in the output of the GST are bounded proportionally to size of the perturbation of the underlying graph. The proportionality constant depends on the model characteristics (number of scales, number of layers, chosen wavelets) but does not depend on characteristics of the graph. Finally, we used numerical experiments to show that the GST representation is also rich enough to achieve comparable performance as the popular GFT, which is a linear, graph-based representation.

#### Acknowledgments

Fernando Gama and Alejandro Ribeiro are supported by NSF CCF 1717120, ARO W911NF1710438, ARL DCIST CRA W911NF-17-2-0181, ISTC-WAS and Intel DevCloud.

Joan Bruna is partially supported by the Alfred P. Sloan Foundation, NSF RI-1816753, NSF CAREER CIF 1845360, and Samsung Electronics.

We thank Edouard Oyallon for reviewing an earlier version of the manuscript and suggesting valuable improvements.

## Footnotes

[1]For notational simplicity, we consider that each node in the graph holds *scalar* data, but the extension to *vector* data is straightforward, see [18, 20] for details.

[2]Datasets and source code: `http://github.com/alelab-upenn/graph-scattering-transforms`

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
