[Supplementary Material · scatteringStability-NeurIPS19-supplementary.pdf]

# Supplementary Materials for 'Stability of Graph Scattering Transforms'

## A    Proof of proposition 1: Permutation invariance

The low-pass summarizing operator $U$ is linear so that, under permutations, $\hat{U} = U\mathbf{P}$. Therefore,

$$\phi_{p_j(\ell)}(\hat{\mathbf{S}}, \hat{\mathbf{x}}) = \hat{U}(\rho\mathbf{H}_j(\hat{\mathbf{S}}))_{p_j(\ell)}\hat{\mathbf{x}} = U\mathbf{P}(\rho\mathbf{H}_j(\mathbf{P}^\mathsf{T}\mathbf{S}\mathbf{P}))_{p_j(\ell)}\mathbf{P}^\mathsf{T}\mathbf{x}. \tag{20}$$

But, for analytical wavelets, we have that $\mathbf{H}_j(\mathbf{P}^\mathsf{T}\mathbf{S}\mathbf{P}) = \mathbf{P}^\mathsf{T}\mathbf{H}_j(\mathbf{S})\mathbf{P}$ [cf. (5)]. Also, since the nonlinearities are pointwise, $\rho(\mathbf{P}^\mathsf{T}\mathbf{z}) = \mathbf{P}^\mathsf{T}\rho(\mathbf{z})$ for any $\mathbf{z}$. Then, we get

$$\phi_{p_j(\ell)}(\hat{\mathbf{S}}, \hat{\mathbf{x}}) = U\mathbf{P}\mathbf{P}^\mathsf{T}(\rho\mathbf{H}_j(\mathbf{S}))_{p_j(\ell)}\mathbf{P}\mathbf{P}^\mathsf{T}\mathbf{x} = U(\rho\mathbf{H}_j(\mathbf{S}))_{p_j(\ell)}\mathbf{x} = \phi_{p_j(\ell)}(\mathbf{S}, \mathbf{x}) \tag{21}$$

where we used that $\mathbf{P}\mathbf{P}^\mathsf{T} = \mathbf{I}$ by definition of permutation matrix. Since this holds independently of $p_j(\ell)$, then it holds for every scattering coefficient. If it holds for every scattering coefficient, then it hold for the GST $\mathbf{\Phi}(\mathbf{x})$, thereby completing the proof.

## B    Proof of proposition 2: Graph wavelet stability

Without loss of generality assume that $\mathbf{P} = \mathbf{I}$ (alternatively, fix some $\mathbf{P}_0 \in \mathcal{P}$ and redefine $\hat{\mathbf{S}}$ to be equal to $\mathbf{P}_0^\mathsf{T}\hat{\mathbf{S}}\mathbf{P}_0$). Then, we can write $\hat{\mathbf{S}} = \mathbf{S} + \mathbf{E}^\mathsf{H}\mathbf{S} + \mathbf{S}\mathbf{E}$. Observe that, given two arbitrary square matrices $\mathbf{A}$ and $\mathbf{B}$ of the same size, the first order expansion of $(\mathbf{A} + \mathbf{B})^k$ gives

$$(\mathbf{A} + \mathbf{B})^k = \mathbf{A}^k + \sum_{r=0}^{k-1} \mathbf{A}^r \mathbf{B} \mathbf{A}^{k-r-1} + \mathbf{C} \tag{22}$$

with $\mathbf{C}$ such that $\|\mathbf{C}\| \leq \sum_{r=2}^{k} \binom{k}{r} \|\mathbf{B}\|^r \|\mathbf{A}\|^{k-r}$. Exploiting that the graph wavelets are analytic functions, we can use this first-order approximation in (5) with $\mathbf{A} = \mathbf{S}$ and $\mathbf{B} = \mathbf{E}^\mathsf{H}\mathbf{S} + \mathbf{S}\mathbf{E}$, to get

$$\mathbf{H}(\hat{\mathbf{S}}) - \mathbf{H}(\mathbf{S}) = \sum_{k=0}^{\infty} h_k \sum_{r=0}^{k-1} \left(\mathbf{S}^r \mathbf{E}^\mathsf{H} \mathbf{S}^{k-r} + \mathbf{S}^{r+1} \mathbf{E} \mathbf{S}^{k-r-1}\right) + \mathbf{D} \tag{23}$$

with $\mathbf{D}$ such that $\|\mathbf{D}\| = \mathcal{O}(\|\mathbf{E}\|_2^2)$.

Next, we proceed to compute the output to an graph signal $\mathbf{x}$ with finite energy $\|\mathbf{x}\| < \infty$ which has a GFT given by $\tilde{\mathbf{x}} = [\tilde{x}_1, \ldots, \tilde{x}_N]^\mathsf{T}$ so that

$$\mathbf{x} = \sum_{i=1}^{N} \tilde{x}_i \mathbf{v}_i \tag{24}$$

for $\{\mathbf{v}_i\}_{i=1}^{N}$ the eigenvector basis of the GSO $\mathbf{S}$. Then, we can compute

$$\left[\mathbf{H}(\hat{\mathbf{S}}) - \mathbf{H}(\mathbf{S})\right] \mathbf{x} = \sum_{i=1}^{N} \tilde{x}_i \sum_{k=0}^{\infty} h_k \sum_{r=0}^{k-1} \left(\mathbf{S}^r \mathbf{E}^\mathsf{H} \mathbf{S}^{k-r} + \mathbf{S}^{r+1} \mathbf{E} \mathbf{S}^{k-r-1}\right) \mathbf{v}_i + \sum_{i=1}^{N} \tilde{x}_i \mathbf{D} \mathbf{v}_i \tag{25}$$

Let us consider first the product $\mathbf{S}^{r+1}\mathbf{E}\mathbf{S}^{k-r-1}\mathbf{v}_i$. It is immediate that $\mathbf{S}^{k-r-1}\mathbf{v}_i = \lambda_i^{k-r-1}\mathbf{v}_i$, so we focus on the product

$$\mathbf{E}\mathbf{v}_i = \sum_{n=1}^{N} m_n \mathbf{u}_n \mathbf{u}_n^\mathsf{H} \mathbf{v}_i = m_N \sum_{n=1}^{N} \frac{m_n}{m_N} \mathbf{u}_n \mathbf{u}_n^\mathsf{H} \mathbf{v}_i. \tag{26}$$

The hypothesis that $\|\mathbf{E}/m_N - \mathbf{I}\| \leq \varepsilon$ is equivalent to $1 - \varepsilon \leq m_n/m_N \leq 1 + \varepsilon$ for all $n = 1, \ldots, N$. Then, we can write $m_n/m_N = 1 + \delta_n$ with $|\delta_n| \leq \varepsilon$, which yields

$$\mathbf{E}\mathbf{v}_i = m_N \mathbf{v}_i + m_N \mathbf{w}_i \quad , \quad \mathbf{w}_i = \sum_{n=1}^{N} \delta_n \mathbf{u}_n \mathbf{u}_n^\mathsf{H} \mathbf{v}_i. \tag{27}$$

Note that

$$\|\mathbf{w}_i\| \le \left\| \sum_{n=1}^{N} \delta_n \mathbf{u}_n \mathbf{u}_n^{\mathsf{H}} \right\| \|\mathbf{v}_i\| = \max_{n=1,\dots,N} |\delta_n| \le \varepsilon. \tag{28}$$

Using (27) we get that

$$\mathbf{S}^{r+1} \mathbf{E} \mathbf{S}^{k-r-1} \mathbf{v}_i = m_N \lambda_i^k \mathbf{v}_i + m_N \mathbf{V} \lambda_i^{k-r-1} \mathbf{\Lambda}^{r+1} \mathbf{V}^{\mathsf{H}} \mathbf{w}_i. \tag{29}$$

And this can be used to compute

$$\sum_{k=0}^{\infty} h_k \sum_{r=0}^{k-1} \mathbf{S}^{r+1} \mathbf{E} \mathbf{S}^{k-r-1} \mathbf{v}_i = m_N \sum_{k=0}^{\infty} h_k (k \lambda_i^k \mathbf{v}_i) + m_N \mathbf{V} \left( \sum_{k=0}^{\infty} h_k \sum_{r=0}^{k-1} \lambda_i^{k-r-1} \mathbf{\Lambda}^{r+1} \right) \mathbf{V}^{\mathsf{H}} \mathbf{w}_i$$
$$= m_N \lambda_i h'(\lambda_i) \mathbf{v}_i + m_N \mathbf{V} \mathrm{diag}(\hat{\mathbf{g}}_i) \mathbf{V}^{\mathsf{H}} \mathbf{w}_i \tag{30}$$

where vector $\hat{\mathbf{g}}_i \in \mathbb{R}^N$ is such that

$$[\hat{\mathbf{g}}_i]_j = \sum_{k=0}^{\infty} h_k \sum_{r=0}^{k-1} \lambda_i^{k-r-1} \lambda_j^{r+1}. \tag{31}$$

We note that if $j = i$ then $\lambda_i^{k-r-1} \lambda_j^{r+1} = \lambda_i^k$ and thus $[\hat{\mathbf{g}}_i]_i = \lambda_i h'(\lambda_i)$. For $j \ne i$, on the other hand, noting that $\sum_{r=0}^{k-1} \lambda_i^{k-r-1} \lambda_j^{r+1} = \lambda_j(\lambda_i^k - \lambda_j^k)/(\lambda_i - \lambda_j)$ we have

$$[\hat{\mathbf{g}}_i]_j = \sum_{k=0}^{\infty} h_k \lambda_j \frac{\lambda_i^k - \lambda_j^k}{\lambda_i - \lambda_j} = \frac{\lambda_j}{\lambda_i - \lambda_j} \sum_{k=0}^{\infty} h_k \left( \lambda_i^k - \lambda_j^k \right). \tag{32}$$

Therefore,

$$[\hat{\mathbf{g}}_i]_j = \begin{cases} \lambda_i h'(\lambda_i) & \text{if } j = i \\ \lambda_j \frac{h(\lambda_i) - h(\lambda_j)}{\lambda_i - \lambda_j} & \text{if } j \ne i \end{cases} \tag{33}$$

We also observe that $|[\hat{\mathbf{g}}_i]_j| \le G \lesssim \max\{C, 2B\}$ for all $j = 1, \dots, N$ due to the fact that $|h(\lambda)| \le B$ and $|\lambda h'(\lambda)| \le C$ due to the integral Lipschitz constraint.

We can get an expression analogous to (30) for the term

$$\sum_{k=0}^{\infty} h_k \sum_{r=0}^{k-1} \mathbf{S}^r \mathbf{E}^{\mathsf{H}} \mathbf{S}^{k-r} \mathbf{v}_i = m_N \lambda_i h'(\lambda_i) \mathbf{v}_i + m_N \mathbf{V} \mathrm{diag}(\check{\mathbf{g}}_i) \mathbf{V}^{\mathsf{H}} \mathbf{w}_i \tag{34}$$

where now

$$[\check{\mathbf{g}}_i]_j = \begin{cases} \lambda_i h'(\lambda_i) & \text{if } j = i \\ \lambda_i \frac{h(\lambda_i) - h(\lambda_j)}{\lambda_i - \lambda_j} & \text{if } j \ne i \end{cases} \tag{35}$$

where it also holds that $|[\check{\mathbf{g}}_i]_j| \le G$.

Finally, using (30) and (34) back in (25), and applying the norm, we get

$$\left\| \left[ \mathbf{H}(\hat{\mathbf{S}}) - \mathbf{H}(\mathbf{S}) \right] \mathbf{x} \right\| \le \left\| 2m_N \sum_{i=1}^{N} \lambda_i h'(\lambda_i) \tilde{x}_i \mathbf{v}_i \right\| \tag{36}$$

$$+ \left\| m_N \sum_{i=1}^{N} \mathbf{V} \mathrm{diag}(\hat{\mathbf{g}}_i + \check{\mathbf{g}}_i) \mathbf{V}^{\mathsf{H}} (\tilde{x}_i \mathbf{w}_i) \right\| \tag{37}$$

$$+ \|\mathbf{D}\tilde{\mathbf{x}}\|. \tag{38}$$

For the first order term (36) we have

$$\left\| 2m_N \sum_{i=1}^{N} \lambda_i h'(\lambda_i) \tilde{x}_i \mathbf{v}_i \right\|^2 = 4|m_N|^2 \sum_{i=1}^{N} |\lambda_i h'(\lambda_i)|^2 |\tilde{x}_i|^2 \tag{39}$$

since $\{\mathbf{v}_i\}_{i=1}^{N}$ form an orthonormal basis. Then, bounding $|m_N| \leq \varepsilon/2$ in virtue of $d(\mathbf{S}, \hat{\mathbf{s}}) \leq \varepsilon/2$ and $|\lambda h'(\lambda)| \leq C$ for all $\lambda$, we get

$$4|m_N|^2 \sum_{i=1}^{N} |\lambda_i h'(\lambda_i)|^2 |\tilde{x}_i|^2 \leq \varepsilon^2 C^2 \sum_{i=1}^{N} |\tilde{x}_i|^2 = \varepsilon^2 C^2 \|\mathbf{x}\|^2. \tag{40}$$

For the second order term (37) coming from $\mathbf{E}\mathbf{v}_i$, we have

$$\left\| m_N \sum_{i=1}^{N} \mathbf{V}\mathrm{diag}(\hat{\mathbf{g}}_i + \breve{\mathbf{g}}_i)\mathbf{V}^{\mathsf{H}}(\tilde{x}_i\mathbf{w}_i) \right\| \leq |m_N| \sum_{i=1}^{N} \|\mathbf{V}\mathrm{diag}(\hat{\mathbf{g}}_i + \breve{\mathbf{g}}_i)\mathbf{V}^{\mathsf{H}}\| |\tilde{x}_i| \|\mathbf{w}_i\| \tag{41}$$

where, by bounding $|m_N| \leq \varepsilon/2$, $\|\mathbf{V}\mathrm{diag}(\hat{\mathbf{g}}_i+\breve{\mathbf{g}}_i)\mathbf{V}^{\mathsf{H}}\| \leq 2G$ in virtue of (33) and (35), $\sum_{i=1}^{N} |\tilde{x}_i| = \|\tilde{\mathbf{x}}\|_1 \leq \sqrt{N}\|\tilde{\mathbf{x}}\|_2 = \sqrt{N}\|\mathbf{x}\|$ and $\|\mathbf{w}_i\| \leq \varepsilon$ because of (28), we get

$$\left\| m_N \sum_{i=1}^{N} \mathbf{V}\mathrm{diag}(\hat{\mathbf{g}}_i + \breve{\mathbf{g}}_i)\mathbf{V}^{\mathsf{H}}(\tilde{x}_i\mathbf{w}_i) \right\| \leq \mathcal{O}(\varepsilon^2)\|\mathbf{x}\|. \tag{42}$$

Finally, for the second order term (38) stemming from the expansion of $\hat{\mathbf{S}}^k$, we obtain

$$\|\mathbf{D}\tilde{\mathbf{x}}\| \leq \mathcal{O}(\|\mathbf{E}\|^2)\|\mathbf{x}\| \leq \mathcal{O}(\varepsilon^2)\|\mathbf{x}\|. \tag{43}$$

Using bounds (40), (42) and (43) back in (36), (37) and (38), respectively, we complete the proof.

## C    Proof of proposition 3: GST coefficient stability

We prove a more general case in which the low-pass average operator $U$ depends on the GSO $\mathbf{S}$ and is such that $\|U\| \leq B_U$ and $\|U(\mathbf{S}) - U(\hat{\mathbf{S}})\| \leq \varepsilon_U$. Prop. 3 can be readily obtained from Prop. 4 below by setting $B_U = 1$ and $\varepsilon_U = 0$ which is the case for the selected low-pass average operator $U = N^{-1}\mathbf{1}^{\mathsf{T}}$, that does not depend on $\mathbf{S}$.

**Proposition 4** (GST coefficient stability). *Let $\mathcal{G}$ be a graph with GSO $\mathbf{S}$ and $\widehat{\mathcal{G}}$ be the* perturbed *graph with GSO $\hat{\mathbf{S}}$, such that $d(\mathbf{S}, \hat{\mathbf{S}}) \leq \varepsilon/2$. Let $\mathbf{E} \in \mathcal{E}(\mathbf{S}, \hat{\mathbf{S}})$, consider its eigendecomposition $\mathbf{E} = \mathbf{U}\mathbf{M}\mathbf{U}^{\mathsf{H}}$ where the eigenvalues in $\mathbf{M} = \mathrm{diag}(m_1, \ldots, m_N)$ are ordered such that $|m_1| \leq \cdots \leq |m_N|$, and assume that the structural constraint $\|\mathbf{E}/m_N - \mathbf{I}\| \leq \varepsilon$ holds. Consider a GST with $L$ layers and $J$ wavelet scales $h_j(\lambda)$, each of which satisfies the integral Lipschitz constraint $|\lambda h_j'(\lambda)| \leq C$ and conform a frame with bounds $0 < A \leq B$ [cf. (9)]. Then, for the coefficient $\phi_{p_j(\ell)}$ associated to path $p_j(\ell) = (j_1, \ldots, j_\ell)$ it holds that*

$$|\phi_{p_j(\ell)}(\mathbf{S}, \mathbf{x}) - \phi_{p_j(\ell)}(\hat{\mathbf{S}}, \mathbf{x})| \leq \left( \varepsilon_U B^\ell + B_U \varepsilon C \ell B^{\ell-1} \right) \|\mathbf{x}\| \tag{44}$$

Starting with (2), using graph convolutions (5) and recalling that we can write $\mathbf{h}_j *_{\mathbf{S}} \mathbf{x} = \mathbf{H}_j(\mathbf{S})\mathbf{x}$, we get

$$\begin{aligned} \left| \phi_{p_j(\ell)}(\mathbf{S}, \mathbf{x}) - \phi_{p_j(\ell)}(\hat{\mathbf{S}}, \mathbf{x}) \right| &= \left| U(\mathbf{S})(\rho\mathbf{H}_j(\mathbf{S}))_{p_j(\ell)}\mathbf{x} - U(\hat{\mathbf{S}})(\rho\mathbf{H}_j(\hat{\mathbf{S}}))_{p_j(\ell)}\mathbf{x} \right| \\ &\leq \left| U(\mathbf{S})(\rho\mathbf{H}_j(\mathbf{S}))_{p_j(\ell)}\mathbf{x} - U(\hat{\mathbf{S}})(\rho\mathbf{H}_j(\mathbf{S}))_{p_j(\ell)}\mathbf{x} \right| \\ &\quad + \left| U(\hat{\mathbf{S}})(\rho\mathbf{H}_j(\mathbf{S}))_{p_j(\ell)}\mathbf{x} - U(\hat{\mathbf{S}})(\rho\mathbf{H}_j(\hat{\mathbf{S}}))_{p_j(\ell)}\mathbf{x} \right| \end{aligned} \tag{45}$$

where we have added and subtracted $U(\hat{\mathbf{S}})(\rho h_j(\mathbf{S}))_{p_j(\ell)}\mathbf{x}$, and then applied the triangle inequality. Applying Cauchy-Schwarz inequality to each term, we get

$$\begin{aligned} \left| \phi_{p_j(\ell)}(\mathbf{S}, \mathbf{x}) - \phi_{p_j(\ell)}(\hat{\mathbf{S}}, \mathbf{x}) \right| &\leq \|U(\mathbf{S}) - U(\hat{\mathbf{S}})\| \|(\rho\mathbf{H}_j(\mathbf{S}))_{p_j(\ell)}\mathbf{x}\| \\ &\quad + \|U(\hat{\mathbf{S}})\| \|(\rho\mathbf{H}_j(\mathbf{S}))_{p_j(\ell)}\mathbf{x} - (\rho\mathbf{H}_j(\hat{\mathbf{S}}))_{p_j(\ell)}\mathbf{x}\|. \end{aligned} \tag{46}$$

We proceed by bounding, one by one, these four terms. The first one, is bounded by hypothesis

$$\|U(\mathbf{S}) - U(\hat{\mathbf{S}})\| \leq \varepsilon_U. \tag{47}$$

A3

For the second term, we recall that the nonlinearity is nonexpansive, i.e. $\|\rho\| \leq 1$, and use the definition of operator norm together with the property of submultiplicativity

$$\|(\rho\mathbf{H}_j(\mathbf{S}))_{p_j(\ell)}\mathbf{x}\| \leq \|\rho\mathbf{H}_{j_\ell}(\mathbf{S})\| \cdots \|\rho\mathbf{H}_{j_1}(\mathbf{S})\|\|\mathbf{x}\| \tag{48}$$

and thus, together with the frame condition (9), we obtain

$$\|(\rho\mathbf{H}_j(\mathbf{S}))_{p_j(\ell)}\mathbf{x}\| \leq B^\ell\|\mathbf{x}\|. \tag{49}$$

The third term is bounded by the hypothesis that the summarizing linear operator is bounded

$$\|U(\hat{\mathbf{S}})\| \leq B_U. \tag{50}$$

The fourth and last term is slightly more involved. We can bound it in a recursive fashion as follows. First, add and subtract $\rho\mathbf{H}_{j_\ell}(\mathbf{S})\rho\mathbf{H}_{j_{\ell-1}}(\hat{\mathbf{S}}) \cdots \rho\mathbf{H}_{j_1}(\hat{\mathbf{S}})$ and use the triangle inequality to obtain

$$\|(\rho\mathbf{H}_j(\mathbf{S}))_{p_j(\ell)}\mathbf{x} - (\rho\mathbf{H}_j(\hat{\mathbf{S}}))_{p_j(\ell)}\mathbf{x}\|$$
$$\leq \|\rho\mathbf{H}_{j_\ell}(\mathbf{S})\left(\rho\mathbf{H}_{j_{\ell-1}}(\mathbf{S}) \cdots \rho\mathbf{H}_{j_1}(\mathbf{S}) - \rho\mathbf{H}_{j_{\ell-1}}(\hat{\mathbf{S}}) \cdots \rho\mathbf{H}_{j_1}(\hat{\mathbf{S}})\right)\|$$
$$+ \|\left(\rho\mathbf{H}_{j_\ell}(\mathbf{S}) - \rho\mathbf{H}_{j_\ell}(\hat{\mathbf{S}})\right)\rho\mathbf{H}_{j_{\ell-1}}(\hat{\mathbf{S}}) \cdots \rho\mathbf{H}_{j_1}(\hat{\mathbf{S}})\|. \tag{51}$$

Now, using submultiplicativity and defining

$$\text{bound}(\ell) = \|\rho\mathbf{H}_{j_\ell}(\mathbf{S}) \cdots \rho\mathbf{H}_{j_1}(\mathbf{S}) - \rho\mathbf{H}_{j_\ell}(\hat{\mathbf{S}}) \cdots \rho\mathbf{H}_{j_1}(\hat{\mathbf{S}})\| \tag{52}$$

we observe that (51) becomes the recursive inequality

$$\text{bound}(\ell) \leq B\,\text{bound}(\ell-1) + \varepsilon CB^{\ell-1} \tag{53}$$

where we have used that $\|\rho\mathbf{H}_{j_\ell}(\mathbf{S})\| \leq B$ by the frame condition, that $\|\mathbf{H}_j(\mathbf{S}) - \mathbf{H}_j(\hat{\mathbf{S}})\| \leq \varepsilon C$ due to Prop. 2, and that $\|\rho\mathbf{H}_{j_{\ell-1}}(\hat{\mathbf{S}}) \cdots \rho\mathbf{H}_{j_1}(\hat{\mathbf{S}})\| \leq B^{\ell-1}$ by the same submultiplicativity and frame argument of (49).

Solving the recursive inequality in (53) we reach

$$\text{bound}(\ell) \leq B^{\ell-1}\,\text{bound}(1) + (\ell-1)\varepsilon CB^{\ell-1} \tag{54}$$

and noting that

$$\text{bound}(1) = \|\rho\mathbf{H}_{j_1}(\mathbf{S}) - \rho\mathbf{H}_{j_1}(\hat{\mathbf{S}})\| \leq \varepsilon C \tag{55}$$

by Prop. 2 we finally bound the fourth term (51) by

$$\|(\rho\mathbf{H}_j(\mathbf{S}))_{p_j(\ell)}\mathbf{x} - (\rho\mathbf{H}_j(\hat{\mathbf{S}}))_{p_j(\ell)}\mathbf{x}\| \leq \ell\varepsilon CB^{\ell-1} \tag{56}$$

Finally, substituting (47), (49), (50) and (56) back in (46), we complete the proof.

## D Proof of theorem 1: GST stability

In this case, we also prove a more general case for a low-pass average operator $U$ that depends on the GSO $\mathbf{S}$ and is such that $\|U\| \leq B_U$ and $\|U(\mathbf{S}) - U(\hat{\mathbf{S}})\| \leq \varepsilon_U$. Theorem 1 can be readily obtained from Theorem 1 below by setting $B_U = 1$ and $\varepsilon_U = 0$ which is the case for the selected low-pass average operator $U = N^{-1}\mathbf{1}^\mathsf{T}$, that does not depend on $\mathbf{S}$.

**Theorem 2** (GST stability). *Let $\mathcal{G}$ be a graph with GSO $\mathbf{S}$ and $\hat{\mathcal{G}}$ be the perturbed graph with GSO $\hat{\mathbf{S}}$, such that $d(\mathbf{S}, \hat{\mathbf{S}}) \leq \varepsilon/2$. Let $\mathbf{E} \in \mathcal{E}(\mathbf{S}, \hat{\mathbf{S}})$, consider its eigendecomposition $\mathbf{E} = \mathbf{UMU}^\mathsf{H}$ where the eigenvalues in $\mathbf{M} = \text{diag}(m_1, \ldots, m_N)$ are ordered such that $|m_1| \leq \cdots \leq |m_N|$, and assume that the structural constraint $\|\mathbf{E}/m_N - \mathbf{I}\| \leq \varepsilon$ holds. Consider a GST with $L$ layers and $J$ wavelet scales $h_j(\lambda)$, each of which satisfies the integral Lipschitz constraint $|\lambda h'_j(\lambda)| \leq C$ and conform a frame with bounds $0 < A \leq B$ [cf. (9)]. Then, it holds that*

$$\left\|\mathbf{\Phi}(\mathbf{S}, \mathbf{x}) - \mathbf{\Phi}(\hat{\mathbf{S}}, \mathbf{x})\right\| \leq \left[\varepsilon_U^2\xi_{BJL}^{(0)} + 2\varepsilon_U B_U\frac{\varepsilon C}{B}\xi_{BJL}^{(1)} + B_U^2\left(\frac{\varepsilon C}{B}\right)^2\xi_{BJL}^{(2)}\right]^{1/2}\|\mathbf{x}\| \tag{57}$$

*with $\xi_{BJL}^{(r)} = \sum_{\ell=0}^{L-1}\ell^r(B^2 J)^\ell$.*

From (3), we get

$$\|\mathbf{\Phi}(\mathbf{S}, \mathbf{x}) - \mathbf{\Phi}(\hat{\mathbf{S}}, \mathbf{x})\|^2 = \sum_{\ell=0}^{L-1} \sum_{j=1}^{J^\ell} |\phi_{p_j(\ell)}(\mathbf{S}, \mathbf{x}) - \phi_{p_j(\ell)}(\hat{\mathbf{S}}, \mathbf{x})|^2. \tag{58}$$

Now, each term in the sum, can be bounded by means of Prop. 3, so that

$$\|\mathbf{\Phi}(\mathbf{S}, \mathbf{x}) - \mathbf{\Phi}(\hat{\mathbf{S}}, \mathbf{x})\|^2 \leq \sum_{\ell=0}^{L-1} \sum_{j=1}^{J^\ell} \left(\varepsilon_U B^\ell \|\mathbf{x}\| + B_U \varepsilon C \ell B^{\ell-1} \|\mathbf{x}\|\right)^2. \tag{59}$$

Expanding the square, and taking $\|\mathbf{x}\|^2$ out of the sum, yields

$$\|\mathbf{\Phi}(\mathbf{S}, \mathbf{x}) - \mathbf{\Phi}(\hat{\mathbf{S}}, \mathbf{x})\|^2 \leq \|\mathbf{x}\|^2 \sum_{\ell=0}^{L-1} \sum_{j=1}^{J^\ell} \left(\varepsilon_U^2 B^{2\ell} + 2\varepsilon_U B_U \varepsilon C \ell B^{2\ell-1} + B_U^2 \varepsilon^2 C^2 \ell^2 B^{2(\ell-1)}\right). \tag{60}$$

We note that no term in the inner sum depends on $j$, so we obtain

$$\|\mathbf{\Phi}(\mathbf{S}, \mathbf{x}) - \mathbf{\Phi}(\hat{\mathbf{S}}, \mathbf{x})\|^2 \leq \|\mathbf{x}\|^2 \sum_{\ell=0}^{L-1} J^\ell \left(\varepsilon_U^2 B^{2\ell} + 2\varepsilon_U B_U \varepsilon C \ell B^{2\ell-1} + B_U^2 \varepsilon^2 C^2 \ell^2 B^{2(\ell-1)}\right)$$

$$\leq \|\mathbf{x}\|^2 \sum_{\ell=0}^{L-1} \left(\varepsilon_U^2 (JB^2)^\ell + 2\varepsilon_U B_U(\varepsilon C/B)\ell(JB^2)^\ell + B_U^2 (\varepsilon C/B)^2 \ell^2 (JB^2)^\ell\right). \tag{61}$$

Assuming $JB^2 \neq 1$, we can use the geometric sum to get

$$\|\mathbf{\Phi}(\mathbf{S}, \mathbf{x}) - \mathbf{\Phi}(\hat{\mathbf{S}}, \mathbf{x})\|^2 \leq \|\mathbf{x}\|^2 \left[\varepsilon_U^2 \xi_{BJL}^{(0)} + 2\varepsilon_U B_U \frac{\varepsilon C}{B} \xi_{BJL}^{(1)} + B_U^2 \left(\frac{\varepsilon C}{B}\right)^2 \xi_{BJL}^{(2)}\right] \tag{62}$$

with

$$\xi_{BJL}^{(0)} = \sum_{\ell=0}^{L-1} (B^2 J)^\ell = \frac{(B^2 J)^L - 1}{B^2 J - 1} \tag{63}$$

$$\xi_{BJL}^{(1)} = \sum_{\ell=0}^{L-1} \ell(B^2 J)^\ell = \frac{B^2 J + (L-1)(B^2 J)^{L-1} - L(B^2 J)^L}{(B^2 j - 1)^2} \tag{64}$$

$$\xi_{BJL}^{(2)} = \sum_{\ell=0}^{L-1} \ell^2 (B^2 J)^\ell \tag{65}$$

$$= \frac{(1 + 2L - 2L^2)(B^2 J)^{L+1} + L^2 (B^2 J)^L + (L-1)^2 (B^2 J)^{L+2} - (B^2 J)^2 - (BJ)^2}{(B^2 J - 1)^3}$$

Finally, we apply the square root to complete the proof.

## E   Details on numerical experiments

Experiment E.1 is a synthetic experiment where we can exercise full control on the perturbation size $\varepsilon$ [cf. (15)]. The objective is to show how stable is the GST when compared to the GFT, and also to show how tight the bound is. Experiments E.2 and E.3 are based on real-world data, in problem formulations analogous to [22]. The objective is to show that the GST is a useful representation, yielding similar performance than the GFT (i.e. that they capture, at least, as rich information as the GFT). Additionally, we show how stable the GST is to real-world perturbations (i.e. perturbations that are not synthetically controlled by fixing $\varepsilon$).

We consider three different GSTs. In all cases, we consider $J = 6$ scales and $L = 3$ layers yielding 43 coefficients. First, we consider the use of a monic cubic polynomial as the generating kernel, see [31, eq. (65)] and ensuing discussion for details. We set $x_1$ to be $\lambda_{\lfloor N/4 \rfloor}$ and $x_2 = \lambda_{\lceil 3N/4 \rceil}$ (i.e.

for the eigenvalues in increasing order, the first fourth of the eigenvalues are affected by the monic polynomial $x_1^{-\alpha} x^\alpha$, and for the last fourth, by $x_2^\beta x^{-\beta}$). The values of $\alpha = \beta = 2$ and $K = 20$ are the same as in [31]. The cubic polynomial for the eigenvalues located between $x_1$ and $x_2$ is designed so that the Kernel has continuous first derivatives. The adopted GSO for the monic cubic polynomial GST is the normalized Laplacian $\mathbf{S} = \mathbf{D}^{-1/2}(\mathbf{D} - \mathbf{W})\mathbf{D}^{-1/2}$ with $\mathbf{D} = \text{diag}(\mathbf{W1})$ the degree matrix and $\mathbf{W}$ the adjacency matrix, as suggested in [31]. We denote this GST as $\boldsymbol{\Phi}_{\text{MC}}(\mathbf{S}, \mathbf{x})$ and refer to it as "Monic Cubic". We also use this "Monic Cubic" polynomial to compute the theoretical value of the bound as in (19). The values of both $B$ and $C$ are obtained numerically by computing $B = \max_{\{\lambda_n\}} |h(\lambda_n)|$ and $C = \max_{\{\lambda_n\}} |\lambda_n h'(\lambda_n)|$ for $\{\lambda_n\}$ the set of eigenvalues of $\mathbf{S}$.

Second, we employ a tight Hann wavelet kernel, see [32, Example 1] for details. More specifically, we implement a generating kernel in [32, eq. (9)] with $K = 1$, $a_0 = a_1 = 1/2$, and $R = 3$. We do warping as suggested in [32, Sec. IV] with a warping function $\omega(\lambda) = \log(\lambda)$. We then construct the remaining wavelets from the generating kernel as in [32, eq. (12)] with a scaling function given by [32, eq. (13)]. The adopted GSO is also the normalized Laplacian. We denote this GST as $\boldsymbol{\Phi}_{\text{TH}}(\mathbf{S}, \mathbf{x})$ and refer to it as "Tight Hann".

Third, we compare with the graph geometric scattering of [28]. In this case, each wavelet is obtained as $\mathbf{H}_j = \mathbf{P}^{2^{j-1}}(\mathbf{I} - \mathbf{P}^{2^{j-1}})$, $j = 1, \ldots, J$ for $\mathbf{P} = 1/2(\mathbf{I} + \mathbf{WD}^{-1})$ the lazy random walk operator, which we adopt as the GSO. The low-pass average operator $U$ in this case comprises of the set of $Q$ moments $[U\mathbf{x}]_q = \sum_{n=1}^{N}[\mathbf{x}]_n^q$ for $q = 1, \ldots, Q$. We adopt $Q = 4$ as in [28]. Note that this increases the number of GST coefficients to 172. We denote this GST as $\boldsymbol{\Phi}_{\text{G}}(\mathbf{S}, \mathbf{x})$ and refer to it as "Geometric".

To compare the stability of the graph-based representation given by the GST, we construct another graph-based representation, namely, the graph Fourier transform (GFT). Given a GSO $\mathbf{S} = \mathbf{V\Lambda V}^{\mathsf{H}}$ and a graph signal $\mathbf{x}$, the GFT is computed as $\tilde{\mathbf{x}}(\mathbf{S}, \mathbf{x}) = \mathbf{V}^{\mathsf{H}}\mathbf{x}$, where the dependence on $\mathbf{S}$ comes through $\mathbf{V}$. We choose the normalized Laplacian as the GSO for computing the eigenbasis $\mathbf{V}$. We note that, unlike the GST, the number of coefficients in the GFT representation is $N$. Therefore, for fair comparison, in experiments E.2 and E.3 we select a number of GFT coefficients equal to the number of GST coefficients. We denote the GFT as $\tilde{\mathbf{x}}(\mathbf{S}, \mathbf{x})$ and refer to it as "GFT".

Additionally, we consider a trainable GIN neural network [36] on the adjacency matrix normalized by the largest eigenvalue. We consider a single-layer that outputs 43 features and ReLU activation functions, followed by a MLP that maps the 43 features per node to a vector of size 2 that we consider to be the logits for the 2 classes in each of the classification problems. We train this network over the same training set used to train the SVM classifier, using a cross-entropy loss function for 40 epochs with batch size of 5 in E.2 and 20 in E.3. In both cases we use an ADAM optimizer with learning rate 0.001 and forgetting factors $\beta_1 = 0.9$ and $\beta_2 = 0.999$. We denote the GIN as $\boldsymbol{\Phi}_{\text{GIN}}(\mathbf{S}, \mathbf{x})$.

### E.1 Relative representation error: Small world graphs

In this first experiment, we consider a small world graph of $N = 100$ nodes, generated randomly by using an edge probability $p_{\text{SW}} = 0.5$ and a rewiring probability $q_{\text{SW}} = 0.1$. We then consider a white noise signal $\mathbf{x}$ with power $\sigma_x^2$ defined on top of this graph, and compute the corresponding representations $\boldsymbol{\Phi}(\mathbf{S}, \mathbf{x})$ for all three GSTs and $\tilde{\mathbf{x}}(\mathbf{S}, \mathbf{x})$ for the GFT. We consider perturbations of the adjacency matrix $\mathbf{W}$ given by $\hat{\mathbf{W}} = \mathbf{W} + \mathbf{E}^{\mathsf{H}}\mathbf{W} + \mathbf{WE}$ [cf. (14)], where error matrix $\mathbf{E} = \text{diag}(\mathbf{e})$ is a diagonal matrix with $\mathbf{e}$ being uniformly random, chosen such that $\|\mathbf{E}\| \leq \varepsilon/2$ and $\|\mathbf{E}/e_{\max} - \mathbf{I}\| \leq \varepsilon$ for $e_{\max} = \text{sign}\{\arg\max |[\mathbf{e}]_n|\} \max |[\mathbf{e}]_n|$. Such a deformation amounts for a local dilation of the edge weights (i.e. the edge weights of the neighborhood of each node are dilated by different values). We control the value of $\varepsilon$ as a parametric sweep from $0.1 \cdot \sigma_x^2$ to $1 \cdot \sigma_x^2$.

To account for the different sources of randomness, we generate 10 random connected graph realizations of the small world model, and for each of these 10 graphs we sweep for 10 different values of $\varepsilon$ linearly spaced. For each value of $\varepsilon$ we consider 10 random realizations of the error matrix $\mathbf{E}$, and for each of these perturbations, we simulate 1000 test signals $\mathbf{x}$ assumed to be white gaussian with zero-mean and power $\sigma_x^2 = 1$. We compute $\boldsymbol{\Phi}(\mathbf{S}, \mathbf{x})$ and $\boldsymbol{\Phi}(\hat{\mathbf{S}}, \mathbf{x})$ for each of the three different GSTs and also $\tilde{\mathbf{x}}(\mathbf{S}, \mathbf{x})$ and $\tilde{\mathbf{x}}(\hat{\mathbf{S}}, \mathbf{x})$ for the GFT. We calculate $\|\boldsymbol{\Phi}(\mathbf{S}, \mathbf{x}) - \boldsymbol{\Phi}(\hat{\mathbf{S}}, \mathbf{x})\|/\|\boldsymbol{\Phi}(\mathbf{S}, \mathbf{x})\|$ for each of the signal and average across all 1000 test signals, and then average these means across all 10 random realizations of the error matrix to obtain an estimate of the relative representation error for

each graph, for each value of $\varepsilon$. We proceed analogously for the GFT. For each value of $\varepsilon$ we get 10 estimates fo the relative representation error, one for each random graph realization. We average these across the 10 graphs and plot them as solid lines of Fig. 2a. We estimate the standard deviation across the 10 graphs and plot them as the error bars. We also show, in dashed line, the value of the bound (19) for the GST using the monic cubic polynomial wavelets (we choose this one due to its simplicity in computing the frame bound $B$ and the integral Lipschitz constant $C$).

## E.2 Authorship attribution: Jane Austen

In this experiment, we consider the problem of authorship attribution. The objective is that, given a text excerpt, we can accurately attribute it to a given author. In particular, we consider works authored by Jane Austen, in the same setting as in [22]. To cast this problem as a graph signal classification problem, we proceed as follows. Given a training set of text excerpts (i.e. text excerpts that we know have been authored by Jane Austen), we build a word adjacency network (WAN) using functional words (i.e. words without semantic meaning such as connectors) by determining their relative positioning in the text. It has been noted that the relative positioning of functional words offers a stylometric signature of the author, see [37] for details. Once the graph is built with $N$ functional words, we ensure it is connected and make it undirected by recomputing the edge weights to be the average of the incoming and outgoing edge weights. Each of these functional words act as a node in the network. We can then associate a graph signal (on top of this WAN graph) to each text by counting the frequency of appearance of the functional words. It is then expected that if the frequency of functional words bears strong relation with the graph, then the given text was written by the author for which the WAN was built.

We consider $N = 224$ function words, and a corpus of 771 text fragments (of approximately $1,000$ words) authored by Jane Austen. We split at random this corpus in training, validation and test sets, and use the training set to build the WAN. It is important to note that the texts included in the training set are the only ones used to build the WAN graph, and therefore, the graph is different depending on what texts were selected for the training set. This is a realistic scenario that models the perturbation in the underlying support arising from an estimation of the graph topology (i.e. we do not know the specific graph topology, but estimate it from data, and therefore the true graph topology might be different from the one we are actually using). We consider 10 different split ratios ranging linearly from $0.2$ to $0.9$ which implies that the number of texts used to build the WAN varies from $154$ to $694$.

Once we build the WAN graph, we use a linear SVM to classify the graph signals. To train this SVM we use the same texts included in the training set (labeled as 1 since they were written by the author of interest), and we add to the training set an equivalent number of texts written by other contemporary authors (labeled as 0 since they were written by other authors, such as Emily Brontë, Edgar Allan Poe, Charles Dickens, among many others). We compute the relevant representation $\mathbf{\Phi}$ and $\tilde{\mathbf{x}}$ for each text in the training set and use this representation to train 4 different SVMs (one for each of the three GSTs and one for the GFT). We also train the GIN on this same training set. To use representations of same size, we consider the GFT to project the signal only on the first 43 eigenvectors of the GSO (low-pass filter). We then build a test set with the remaining texts by the author (those not used to build the WAN nor to train the SVM) and add an equivalent number of texts by other contemporary authors. The classification accuracy at test time for different sizes of training set is shown in Fig. 2b.

To account for randomness, for each split ratio (i.e. for each total number of training samples), we simulate 10 different data splits. We compute the classification accuracy averaged over the test set for each of these 10 different data splits. By averaging over the 10 data splits we obtain the mean classification accuracy showed in solid lines in Fig. 2b. We also include the standard deviation estimated from these 10 data splits. While the richness of the GST representation can be observed by the fact that the classification accuracy is comparable to that achieved by using the GFT representation, we can also observe the stability of the GST when compared to the GFT. More specifically, we see (i) that the mean value of the classification accuracy of the GFT oscillates much more than the mean value of the GST, and (ii) that the error bars for the GFT are much larger than those for the GST. This shows that, depending on how we build the underlying graph (i.e. which and how many texts we use to estimate the WAN), the classification accuracy by using the GFT representation can vary wildly. We also note that the trained GIN exhibits around 5 percentage points increase in accuracy with respect to the best performing GST.

### E.3    Source localization: Facebook subnetwork

For the final example, we consider a source localization problem (synthetic data) on a Facebook subnetwork of $N = 234$ users (real-world data), the same as in [22]. This Facebook graph, exhibiting a two-community topology, is a subnetwork of the larger $4,039$ user graph provided in [39]. The problem of source localization consists in observing a diffusion signal and pinpointing to where it started. In the context of graph signals, we consider a signal $\boldsymbol{\delta}_c$ which is a signal with a 1 at node $c$ and 0 elsewhere. Then, we observed the diffused signal $\mathbf{x} = \mathbf{W}^t \boldsymbol{\delta}_c$ for some unknown time $t < t_{\max}$ and we want to estimate the community $c$ that originated the rumor. In a two-community graph, this is a binary classification problem. This problem is analogous to identifying the source of a rumor that spread through the social network.

In this case, we consider perturbations stemming from randomly dropping edges with probability $p$, ranging in 10 logarithmically-spaced points from $0.01$ to $0.3$, as in [22]. This models changing friendships in the network. We use the underlying given graph as $\mathbf{S}$ which we use to build the representations $\boldsymbol{\Phi}(\mathbf{S}, \mathbf{x})$ and $\tilde{\mathbf{x}}(\mathbf{S}, \mathbf{x})$, but we use data $\mathbf{x}$ generated on graph $\hat{\mathbf{S}}$ that corresponds to some random realization of the edge dropping. Again, the objective of this simulation is twofold: (i) to show that using the GST representation achieves as good classification accuracy as using the GFT, and (ii) that the GST is more stable than the GFT. We consider $43$ GFT coefficients belonging to middle frequencies (bandpass filter).

To perform the classification, we train a Linear SVM on the representations obtained for each of the three GSTs, the GFT and the GIN, analogously to experiment E.2. We train the SVMs by generating $1,000$ training samples $\mathbf{x} = \mathbf{W}^t \boldsymbol{\delta}_c$ for $c \in \{c_0, c_1\}$ and random $t < t_{\max} = 20$. The source nodes $c_0$ and $c_1$ are the nodes numbered 38 and 224 since each of them belongs to a different community, and half of the training samples were originated at $c_0$ and the other half at $c_1$. For testing, we generate 200 new samples, half for each community, with random diffusion times $t < t_{\max}$. Results are shown in Fig. 2c.

To account for randomness, we generate 10 different random edge-failing graph realizations, for each value of $p$ simulated. We average across these 10 realizations to obtain the solid lines in Fig. 2c, and compute the standard deviation for the error bars. We observe that $\boldsymbol{\Phi}_{\mathrm{MC}}$, $\boldsymbol{\Phi}_{\mathrm{TH}}$ and $\tilde{\mathbf{x}}$ perform similarly, but that $\boldsymbol{\Phi}_{\mathrm{TH}}$ exhibits considerably less variation and thus is more stable. The geometric scattering $\boldsymbol{\Phi}_{\mathrm{G}}$ exhibits a performance comparable to the GIN $\boldsymbol{\Phi}_{\mathrm{GIN}}$.