[Reviews · NeurIPS 2019]

Reviewer 1



-- Post author-feedback comments -- Following the author responses and suggested improvements, I am raising my score and fully recommend accepting the paper. One minor remark (since this has not been directly addressed in the author feedback): reference [21] in the original submission points to a preprint version of Zou & Lerman "Graph convolutional neural networks via scattering". This paper appears to be in press in Applied and Computational Harmonic Analysis (https://doi.org/10.1016/j.acha.2019.06.003), and therefore this reference should be updated accordingly to point to that in-press journal version. -- Initial review -- This paper explores the stability of graph scattering transforms, which extend the scattering transform that has been studied for some time in traditional spatiotemporal data such as image classification and signal processing. Indeed, the extension of scattering transform has gained attention lately, together with a wider effort to extend the success of convolutional neural networks to network analysis and graph learning. In particular, in the past two months several graph scattering works have appeared, for example, in the last ICLR (Gama et al., 2019), ICML (Gao et al., 2019), and in the journal of applied and computational harmonic analysis (Zou and Lerman, available online June 13, 2019). It should be noted that the mentioned papers have all been published after the submission deadline of NeurIPS, and therefore I would recommend the authors update their introduction and related work sections to reflect the recent interest in this topic and to position their work with respect to recent advances. The results presented in this work do seem to further advance the understanding of graph scattering and one's ability to extend the theoretical foundations of spatiotemporal scattering to graphs and networks. The authors consider here a generic graph scattering transforms that can be realized by choosing appropriate graph wavelets. Similar to Zou and Lerman, they consider invariance to permutations as an equivalent of translations (from traditional signals), and a certain family of perturbations as an equivalent of the deformations considered in traditional scattering studies. The latter result is interesting, as it formulates a new notion of deformation families on graphs, together with an associated notion of stability that does not directly depend on the eigenvalues (or spectral gap) of the associated graph laplacian. The authors also essentially remark that the bound provided by their Theorem 1 is a more natural extension of traditional scattering stability results, as it partially recovers the result from Mallat (2012) when applying the theorem to a line graph, albeit with certain caveats. I believe this point warrants more attention than a minor remark, and I would recommend it be formalized in the main paper in the form of a corollary. However, while the work presented here is clearly timely and the results should be of interest to those familiar with Mallat's work (and related extensions), I am not sure how accessible it would be for a wide audience targeted by NeurIPS. I would imagine the main communities targeted by such work should be the ones looking into network and graph representation learning. If this is the case, it might be difficult for many readers to really understand the impact of the presented bounds and results. For example, how would the error matrix E relate intuitively to the type of operations one might expect to perform when "deforming" a graph? In the traditional case, operations phrased in terms of translations, rotations, scaling, etc. are quite easily perceived. Further, the structural constraint posed in Proposition 2 seems to severely limit the type of operations that are within the scope considered here. This includes some rather simple operations like adding and removing random edges, to which one would certainly want the representations to be stable. These restrictions need further discussion and some intuitive motivating examples of nontrivial perturbations of interest (i.e., beyond just permutations) for which the provided theory applies. It should also be noted that since this work targets the task of comparing graph structures via extracted features, it should properly refer to current notions of graph similarities. A classic task considered in this area is whether two graphs are isomorphic to each other, which gave rise to the Weisfeiler-Lehman test. This in turn stemmed extensive work on WL graph kernels and more recently graph neural networks based on message passing (or message propagation) over networks. The authors should discuss the relations (or differences) between their notion of graph perturbation and the ones considered by such previous work. The last point leads naturally to the main weakness of the paper, which is the somewhat unconvincing numerical evaluation. The experiments presented here do not reflect the vast and extensive amount of previous work done on graph representation and classification tasks. It is disappointing to find the evaluation here does not compare the classification results of the proposed graph scattering to any graph kernel or graph network. Moreover, this evaluation also ignores the many publicly available graph datasets (e.g., from biochemistry data to social networks) that have been used for a while as a standard benchmark for graph classification, both with graph kernels and graph networks. The evaluation here instead uses three somewhat anecdotal and obscure examples. I guess the lack of proper evaluation could be excused in some sense since previous work (cf. Gao et al., ICML 2019) have demonstrated results indicating graph scattering (albeit with somewhat different filters) could be competitive with other established methods on standard benchmark datasets, but if this is the intention here, it should be explicitly stated and properly explained. Even so, the results in figure 2c, discussed in lines 303-309 seem to clearly point to a certain weakness in the graph scattering representation, which the authors attribute (as a "plausible explanation") to the sensitivity of the proposed transform to dropping edges. As mentioned above, this warrants further discussion to clarify the strengths that would balance such weakness, i.e., in terms of which perturbations would not "degrade the stability of the GST". In conclusion, this is a good paper with interesting results, but the presentation should make it more accessible to wider audience and better position it with respect to the current state of the art in graph representation learning.

Reviewer 2



Although the experiments are not sufficient, I like the theoretical analysis part of this work. I have one question in the experimental part: How to obtain the theoretical bound in Fig. 2 (a)? If it is directly computed based on Eq. (19), how to decide the constants B and C? Typos: Line 5: “dons” Eq. (11): “N” -> “N x N”

Reviewer 3



Graph neural networks are receiving an increasing amount of attention, with many new papers appearing in the last few years. Along the way, a subset of these papers are attempting to understand the theoretical underpinnings of such constructions; one such approach, which this paper follows, is by adapting the wavelet scattering transform to graphs. To that end, this paper makes a significant theoretical contribution in proving the stability of graph scattering transforms with a bound that is independent of the graph (maybe, I have question on this below). In doing so, the authors introduce a new perturbation model that measures the distance of a graph perturbation from a permutation, which is a type of graph analogue to how the (regular) scattering transform measures the size of diffeomorphism relative to a translation. This is also significant, as understanding the “right” way to measure graph perturbations is an important part of understanding how to construct graph neural networks and understanding the relevant theoretical properties of them that help explain their performance. The paper is well written and does a good job of introducing the notions of graph convolution, graph wavelets, and the graph scattering transform. Having said that, two areas for possible improvement would be: (1) The notation in Section 3 (before 3.1) is a bit heavy, and I imagine readers not familiar with the scattering transform may need more than one pass to fully absorb everything there; (2) I would have liked to have seen a deeper discussion on how the architecture of the graph scattering transform relates to recent / state-of-the-art graph neural network architectures, beyond the fact that they both (presumably) employ some notion of graph convolution. This would give the reader a better sense for how relevant these results are to other graph neural networks. The numerical experiments are the weakest part of the paper, perhaps justifiably so since the emphasis is on theory. Comparing to the graph Fourier transform is good, but building upon my previous remark relating the construction of the graph scattering transform to other graph neural networks, it would have been nice to see at least one numerical experiment that compares the graph scattering transform with a recent / state-of-the-art graph neural network. On the positive side, it is nice to read the links between the numerical results and the theory presented earlier in the paper. Regarding the theoretical results, I have a few specific questions: - In Prop. 3 / Theorem 1, the integral Lipschitz constraint must be uniform over all j - this seems really strong. It would be nice if the authors could clarify exactly how strong an assumption this is (noting that I see the comment in the paper about the wavelet being constant in the high frequencies, but there it’s just on h and now the prop/thm statements its uniform over all j) - In Thm 3, is the bound really independent of the graph? Indeed, the structural constraint relates E and its largest eigenvalue to epsilon, and E depends on S and \hat{S}, which in turn depend on G and \hat{G}. Furthermore, it seems to me that as N increases, you may want J to increase so that you can have wavelets that go across all length scales in the graph. Anyway, some clarifications would be appreciated. - Does the theory work when G and \hat{G} have different numbers of vertices, as is the case in many tasks? If not, what breaks down, what are the difficulties? Readers would probably be (justifiably) interested in some comments along these lines. Finally, there appear to be a few other graph scattering papers not referenced. I simply point them out, making no insistence on their inclusion: - Xu Chen, Xiuyuan Cheng, Stephane Mallat. “Unsupervised Deep Haar Scattering on Graphs,” Advances in Neural Information Processing Systems 27, pages 1709-1717, 2014 - Dongmian Zou, Gilad Lerman. “Encoding Robust Representation for Graph Generation,” arXiv:1809.10851, 2018 - Feng Gao, Guy Wolf, Matthew Hirn. “Geometric Scattering for Graph Data Analysis,” Proceedings of the 36th International Conference on Machine Learning, PMLR 97, pages 2122-2131, 2019 === Post author feedback === I am mostly pleased with author feedback and will update my score from 7 to 8.

[Author Response · NeurIPS 2019]

We thank all the reviewers and the AC for their time, effort and constructive feedback. [W1], [W2] and [W3] are references included in this response.

**R1, R2, R3:** Suitability of numerical experiments. We appreciate the concern of all reviewers with respect to the numerical simulations. We would like to note that (i) this is mainly a theoretical paper that proves properties of the GST (as R3 remarked) and that (ii) the Diffusion GST is very similar to the method in [W1] which has been compared extensively with other methods (as R1 observed), and therefore we expect similar numerical results as those in [W1]. In any case, we understand, and share, the concerns of the reviewers, so we propose the following changes to the numerical section. First, we will include an explicit comparison with the GST of [W1]. This method will replace the diffusion scattering, since both are very similar (the only differences being the use of the lazy random walk matrix instead of the lazy adjacency matrix, and the use of moments beyond the mean for the low-pass operator $\phi$). Second, we will include comparison with a trainable GIN in [W2], in terms of stability of the resulting architectures. We note that comparing performance with trainable GNNs is tricky since it is highly dependent on the size of the available training set and the details of the training stage (number of epochs, learning rate, etc.), which do not occur in GSTs (which are not trainable). Third, we will add clarification and proper links to the Facebook graph [35] and the authorship attribution dataset [36, W3], to emphasize that these are publicly available, while explaining that we are concerned with datasets involving graph signals, since we want to show how changes in the underlying topology affect the processing of the same signals (i.e. datasets involving graph classification, as those in [W1], where changing the underlying graph changes the graph signal are not useful to illustrate Theorem 1 –even though, in practice, they work–). Fourth, as suggested by R1, we will add a clarification and give due credit to the very good work of [W1] to refer to a more exhaustive comparison between GSTs and other state-of-the-art methods. We hope that these changes will address the concerns of the reviewers. **R1:** Structural constraint and recovery of Mallat's scattering result. The structural constraint allows for edge weights dilations or contractions (i.e. all edge weights increase or all edge weights decrease, albeit with different relative changes). This is required to control the impact that topology changes have on the eigenvectors. Changes such as adding or dropping edges incur in a constant value $\varepsilon = \mathcal{O}(1)$, and as such fix a nonzero minimum for the upper bound. In the very limited number of cases when a topology change can be exactly pinpointed to a change in the eigenvectors, the results in this paper can be improved. One such case is that of the line graph, where it is known how the eigenvectors change when dilating and contracting the edge weights (the effect of a diffeomorphism in [10]), and thus recovering the result in [10]. As space allows, the first observations will be added before Remark 2, while the latter observation will be moved from Remark 1 to a new paragraph and expanded. If necessary, further clarifications on these relationships will be discussed in the supplementary material. **R1:** Graph similarity measures. We would like to clarify that the task is not to compare graph structures in terms of their extracted features, but to analyze how features extracted from graph signals change when the underlying support changes (either because it changes with time, or because it is unknown and has to be estimated, among other examples). The measure of similarity we use in this work is reminiscent of the Gromov-Hausdorff distance, albeit using the spectral norm of the GSO, instead of a max-norm. The comparison with Weisfeller-Lehmann test will hopefully be taken into account by the inclusion of the GIN [W2] in the numerical experiments. **R3:** Relation to other GNNs. Most existing GNNs (with the notable exception of GATs) regularize the linear transform of traditional neural networks by using a graph convolution (5). In this respect, the main computational core of doing a graph convolution followed by pointwise nonlinearities, is the same in GSTs than in GNNs. The main exception, though, is that while GNNs learn the filter coefficients $h_k$ (through different parameterizations), GSTs design them using graph wavelets. Likewise, since Prop. 2 shows stability of the graph filters, which are the same as for GNNs, our stability results may be extended to GNNs with appropriate regularization (since trainable parameters will appear in the bound constants) which is the subject of ongoing work. **R3:** Prop. 3. The formal assumption in Prop. 3 indicates that all involved graph filters in the multirresolution wavelet bank have to satisfy the integral Lipschitz continuity. However, this can be inherited directly from the mother wavelet satisfying the requirement. The hypothesis in Prop. 3 will be changed to reflect this. **R3:** Theorem 1. The bound in Theorem 1 depends on difference between the graphs as defined in (16). This difference will certainly depend on the particularities of the graph topologies considered. Theorem 1 states that it does not depend on the spectral norm of the graph. This will be clarified after (19). **R3:** Different number of nodes. As the theorem is stated now, it requires that both graphs have the same number of nodes. The case when they do not, can be addressed by using correspondences in the same manner as Gromov-Hausdorff distance. This case is beyond the scope of this paper and is currently ongoing work. **R2:** Computation of bound in Fig. 2. The bound in Fig. 2 is computed as in (19). The values of all the constants involved are explained in the supplementary material due to lack of space. In any case, we will add a specific clarification pointing out to this fact in the revised version. **R3:** Update of literature review. We thank the reviewer for bringing to our attention this recently published papers. They will be added to the introduction, and discussed.

[W1] F. Gao, G. Wolf, and M. Hirn, "Geometric scattering for graph data analysis", in *ICML 2019*.

[W2] K. Xu, W. Hu, J. Leskovec, and S. Jegelka, "How powerful are graph neural networks?" in *ICLR 2019*.

[W3] E. Isufi, F. Gama, and A. Ribeiro, "Generalizing Graph Convolutional Neural Networks with Edge-Variant Recursions on Graphs," in *EUSIPCO 2019*.


[Meta-Review · NeurIPS 2019]

A solid, timely and sound paper on the topic of scattering transforms for graphs. Please take into account the remarks of the reviewers in the final version of the paper.